# ^11^C-Choline PET/CT vs. ^99m^Tc-MIBI/^123^Iodide Subtraction SPECT/CT for Preoperative Detection of Abnormal Parathyroid Glands in Primary Hyperparathyroidism: A Prospective, Single-Centre Clinical Trial in 60 Patients

**DOI:** 10.3390/diagnostics10110975

**Published:** 2020-11-19

**Authors:** Afefah Ismail, Julie Wulf Christensen, Martin Krakauer, Susanne Bonnichsen Søndergaard, Bo Zerahn, Birte Nygaard, Finn Noe Bennedbæk, Bent Kristensen, Lars Thorbjørn Jensen

**Affiliations:** 1Department of Nuclear Medicine, Herlev and Gentofte Hospital, 2730 Herlev, Denmark; Afefah.Ismail.02@regionh.dk (A.I.); Martin.Krakauer@regionh.dk (M.K.); Susanne.Bonnichsen.Soendergaard@regionh.dk (S.B.S.); Bo.Zerahn@regionh.dk (B.Z.); Bent.Kristensen.01@regionh.dk (B.K.); Lars.Thorbjoern.Jensen@regionh.dk (L.T.J.); 2Department of Medicine, Division of Endocrinology, Herlev and Gentofte Hospital, 2730 Herlev, Denmark; Birte.Nygaard@regionh.dk (B.N.); Finn.Noe.Bennedbaek@regionh.dk (F.N.B.)

**Keywords:** prospective cohort, primary hyperparathyroidism, method comparison, Choline PET, MIBI-subtraction SPECT, non-inferiority, clinical trial

## Abstract

Background: In patients with primary hyperparathyroidism (PHPT) locating hyperfunctioning glands (HPGs) is crucial when planning minimally invasive surgery. Dual-isotope subtraction scintigraphy with ^99m^Tc-MIBI/^123^Iodide using SPECT/CT and planar pinhole imaging (Method A) has previously shown a sensitivity >93%. However, the method is costly and time consuming and entails a high radiation dose. ^11^C-Choline PET/CT (Method B) is an appealing candidate method unencumbered by these disadvantages. Methods: Sixty patients with newly diagnosed PHPT participated and were scanned using both methods prior to parathyroidectomy. We investigated whether sensitivities of Method A and Method B are similar in a method-to-method comparison when using surgical findings as the true location. Results: At the patient level, sensitivities were (A) 0.98 (95% CI: 0.90–1.00) and (B) 1.00 (95% CI: 0.93–1.00). At the gland level, sensitivities were (A) 0.88 (95% CI: 0.78–0.94) and (B) 0.87 (95% CI: 0.76–0.92). With a non-inferiority margin of ∆ = −0.1, we found a 1-sided *p*-value < 0.001. Conclusion: Our methods comparison study found that sensitivity of Method B was not inferior to Method A. We suggest that ^11^C-Choline PET/CT is a clinically relevant first-choice candidate for preoperative imaging of PHPT and that Method B can likely replace Method A in the near future.

## 1. Introduction

Primary hyperparathyroidism (PHPT) is a relatively common and most often benign disease caused by one or more hyperfunctioning parathyroid glands (HPGs). The disease is characterized by increased ionized calcium (Ca^2+^) and increased or high normal parathyroid hormone (PTH) in plasma. PHPT is frequently detected by chance when blood samples are taken during investigation of uncharacteristic psychiatric, neurological or rheumatic symptoms or if a patient is examined for osteoporosis or kidney stones [1,2,3]. 

Parathyroidectomy (PTx) is currently the only curative treatment for PHPT. In Denmark, the most frequent indications for surgery are plasma Ca^2+^ repeatedly > 1.45 mmol/L or mild hypercalcemia (plasma Ca^2+^ < 1.45 mmol/L) combined with age < 50 years; creatinine clearance < 60 mL/min; BMD T-score < −2.5 in lumbar spine, hip or distal forearm; low-energy fracture; kidney stones or nephrocalcinosis and peptic ulcer. More than 100 new cases of PHPT are diagnosed annually per one million citizens in Denmark, and the highest incidence of PHPT occurs among individuals aged 40–70 years. Minimally invasive surgery is considered as the standard procedure for parathyroidectomy but necessitates correct preoperative localization of the HPGs to guide the surgeon.

Several methods exist for preoperative localization of the HPGs. However, dual-isotope subtraction parathyroid scintigraphy with ^99m^Tc-MIBI and ^123^Iodide including SPECT/CT and planar pinhole imaging has a high diagnostic accuracy and is currently the most widely used method in eastern Denmark. The method has proven to be superior to both single-isotope MIBI planar and SPECT/CT imaging and to dual isotope subtraction scintigraphy with single isotope SPECT/CT [4,5,6,7].

Unfortunately, the method is expensive, time-consuming and somewhat cumbersome and yields a relatively high radiation dose (≈13 mSv) to the patient and, thus, potentially to the staff. Thus, it seems appropriate to investigate alternative methods for the preoperative localization of HPGs. One such candidate is positron emission tomography/computed tomography (PET/CT) using carbon-11-labelled choline (^11^C-Choline) as the tracer, a method that has shown promising results. A recent review and meta-analysis of 14 studies including 517 patients investigated the diagnostic performance of choline PET and concluded that further studies were needed [8]. The same conclusion was drawn in a review showing that the level of evidence for the use of choline PET was medium, 3a based on the Oxford criteria [9]. Though relatively easy to make, 11C-Choline has a short half-life of only 20 min, which is potentially the main challenge of this method. Only facilities with access to a cyclotron laboratory and PET radiochemistry facilities will be capable of performing the ^11^C-Choline PET/CT scans.

The present study aimed to compare the performance (i.e., sensitivity) of ^11^C-choline PET/CT to that of the accepted standard method. The study was a focused “method A to method B” study where the surgical findings served as the reference standard.

We performed a prospective, cohort study comparing the standard method with ^11^C-choline PET/CT. 

## 2. Materials and Methods

### 2.1. Patients

From 19 August 2018 to 24 May 2019, all patients referred to the Department of Medicine, Division of Endocrinology, Herlev and Gentofte Hospital, Denmark, with PHPT were evaluated for inclusion in the present study. The department receives unselected patients from within the Capital Region of Denmark on a strictly administrative basis and not on case severity.

Patients were invited to participate in the study after confirmation of inclusion criteria, i.e., the diagnosis of PHPT and indication for PTx. Exclusion criteria were (1) inability to give informed consent; (2) age < 18 years; (3) inability to cooperate, e.g., in cases of severe claustrophobia; (4) other reasons for secondary osteoporosis (e.g., treatment with glucocorticoids); (5) pregnancy or breastfeeding; (6) known hypothyroidism; (7) known cancer (other than basal cell carcinoma) and (8) allergy to iodine-containing contrast agents.

The study was conducted in accordance with the Helsinki 2 declaration and The International Council for Harmonisation Guideline for Good Clinical Practice (ICH_GCP) clinical trial, approved by the Research Ethics Committee of the Capital Region of Denmark (Journal-nr.:H-18012490, date of approval: 18 June 2018) and the Danish Medicines Agency (EudraCT no. 2018-000726-63, date of approval: 6 June 2018). The GCP-unit in Eastern Denmark has carried out regular monitoring of the trial according to GCP (ID: 2018-1050).

### 2.2. Imaging

#### 2.2.1. ^99m^Tc-MIBI/^123^Iodide Subtraction SPECT/CT and Planar Pinhole

Planar pinhole and SPECT/CT images were acquired on a Philips Skylight gamma-camera (Philips Healthcare, the Netherlands) and a Siemens Symbia Intevo SPECT/CT scanner (Siemens Healthineers, Erlangen, Germany) in the Department of Nuclear Medicine, Herlev and Gentofte Hospital, Denmark. The acquisition details are described elsewhere [4]. Planar subtraction images were analyzed using Oasis software (Segami, Columbia, New York, NY, USA). ^123^I images were subtracted from ^99m^Tc-MIBI images with increasing multiplication until counts in the thyroid bed were equal to ^99m^Tc background counts on the neck. Subtraction SPECT/CT was done by subtracting the reconstructed, attenuation-corrected ^123^I dataset from the ^99^Tc SPECT datasets using MIM (MIM software Inc., Cleveland, OH, USA) in a customized workflow where the subtraction factor could be set and modified at the reader’s discretion. All subtraction planar and SPECT/CT scans were read by the same experienced specialist in nuclear medicine. 

See Figure 1 and Figure 2 for images (same patient as Figure 3).

#### 2.2.2. ^11^C-Choline PET/CT

The order of 11C-Choline PET/CT scans and SPECT/CT scans varied depending on availability. ^11^C-Choline was synthesized in the Cyclotron and Radiopharmaceutical Laboratory at the Department of Nuclear Medicine, Herlev and Gentofte Hospital, Denmark, using the Scansys synthesis module (Scansys Laboratorieteknik, Værløse, Denmark). The injected dose was 400 MBq, and acquisition started 10 min (±5 min) after injection. PET/CT scans were done on a Siemens Biograph with a 64-slice CT scanner (Siemens Healthineers, Erlangen, Germany) in the same department. CT scans were performed in diagnostic quality without contrast. Image processing and description were done according to standard procedures using Syngo.Via (Siemens Healthineers, Germany), Philips Intellispace Portal (Philips Healthcare, Amsterdam, the Netherlands) and AGFA IMPAX software (AGFA Healthcare N.V., Mortsel, Belgium).

All the ^11^C-Choline PET/CT scans were read by the same experienced nuclear medicine specialist. See Figure 3 for images (same patient as Figure 1 and Figure 2).

The specialists were blinded to the results of the competing modality, ensuring unbiasedness as to whether the patient had detectable HPGs.

Both readers noted the number of HPG(s) as well as location relative to the thyroid gland on a coded sheet.

Henceforth, the abbreviations Di-SPECT and choline PET are used instead of ^99m^Tc-MIBI/^123^I-subtraction SPECT/CT planar pinhole and ^11^C-Choline PET/CT.

### 2.3. Surgery

All operating surgeons were head and neck disease specialists and had extensive experience in thyroid/parathyroid gland operations. At the time of surgery, the results of both imaging modalities were available to the surgeon in order to best guide the surgical procedure. Surgical findings served as “true location” of the hyperfunctioning gland to be compared with the results of the scans. Successful surgery was verified by a perioperative plasma PTH decrease of ≥50%. Surgeons recorded the number and location of HPG(s) in the surgical notes.

### 2.4. Statistical Analyses

Statistical analyses were conducted using ‘R’ version 4.0.2 (R Core Team (2020). R: A language and environment for statistical computing. R Foundation for Statistical Computing, Vienna, Austria. URL https://www.R-project.org/). The analyses for effect variables and non-inferiority for clustered matched-pair binary data were carried out according to the recommendations described in detail elsewhere [10,11,12,13,14,15,16,17]. Briefly, biased estimates of sensitivity and specificity may result when positively correlated data within each cluster (i.e., the patient) are ignored. Ignoring this correlation may result in misleadingly small estimated standard errors (and subsequently narrower confidence intervals) for the effect variables because all observations are erroneously counted as independent observations.

As all included patients were diagnosed with PHPT, they had to have at least one true HPG, and therefore, no patient should be a “true negative” with respect to the occurrence of HPG(s). True negatives only occur because our methods are not yet perfect. For this reason, and because the main interest was locating HPGs, our primary effect measures were the sensitivities of the two methods.

Depending on the clinical context and consequences, the analyses of effect variables may be done per patient or using multiple observations per patient (i.e., per gland). For the sake of completeness, we performed analyses for both situations.

Since the standard method already exhibits a high sensitivity (>93%), the aim to demonstrate superiority is nonsensical, and non-inferiority testing was performed instead. As SPECT/CT has previously demonstrated sensitivity greater than 90%, a minimum clinically relevant non-inferiority margin of ∆ = −10% was chosen for this study. This margin is consistent with sensitivity values recommended in other studies. In accordance with recommendations for conducting non-inferiority studies, the confidence interval was set to 90% and a one-sided test was performed with corrections for clustered matched-pair binary data [12].

For all other reported confidence intervals, a level of 95% was chosen and all reported *p*-values are exact.

## 3. Results

Sixty patients met the inclusion criteria and underwent both imaging modalities as well as the standard surgical procedure. The results of the scans were compared to the surgeons’ reported localization of the HPGs. Six patients had two glands removed. See Table 1 for details.

Statistical analyses were performed as a method-to-method comparison of sensitivities at both the patient level, where the surgical findings were tested against the preoperative results of the Di-SPECT and choline PET, respectively, and at the lesion level, i.e., all positive findings count as observations, whether or not more than one HPG was present per patient.

### 3.1. Analysis at the Patient Level

Table 2a,b show results at the patient level for Di-SPECT and choline PET, respectively.

A comparison of the paired data using the cluster corrected McNemar test for marginal homogeneity had a chi-square statistic = 0.089 and a *p*-value = 0.77, showing no differences between the two methods.

### 3.2. Analysis at Parathyroid Gland (“Lesion”) Level

Using a so-called “sandwich estimator” of variance with correlation adjusted confidence intervals, the sensitivity for MIBI-SPECT was 0.83 (95% CI: 0.72–0.91), and for choline PET the sensitivity was 0.87 (95% CI: 0.76–0.93) [11].

Table 3a,b shows results at the gland level for Di-SPECT and choline PET, respectively.

### 3.3. Non-Inferiority Analysis

With a Z statistic = −3.68, a 1-sided *p*-value < 0.001 and a 90% confidence interval ranging from −0.033 to 0.067 (with the lower limit above ∆ = −0.1) the null hypothesis: “choline PET scan is inferior to Di-SPECT” was rejected. Based upon the current data, we found that choline PET was not inferior to the Di-SPECT.

## 4. Discussion

The widely used method in Eastern Denmark for preoperative localization of hyperfunctioning parathyroid glands—the MIBI-I-SPECT subtraction method—has a considerably high sensitivity but is time consuming (≈3 h) and provides a relatively high radiation dose to the patient (≈13 mSv). In most other publications, the SPECT/CT method is based solely on the MIBI wash-out (“dual phase”) acquisition rather than the ^99m^MIBI/123I dual isotope subtraction technique. In contrast, the choline PET scan is completed in under 30 min and provides half the radiation dose (≈6.6 mSv). The long acquisition time for the SPECT/CT requires more staff, and the combination of MIBI and Iodide is more expensive than the production of carbon-11-labelled choline—with the prerequisite of having a cyclotron laboratory in which to produce the tracer.

Most studies on choline PET in the detection of hyperfunctioning parathyroid glands use fluorine-18 rather than carbon-11 although others have used ^11^C-Choline [8,18,19,20,21]. However, there is presumably no difference between the two tracers [21].

The analyses of the two methods applied to a prospective cohort of unselected patients covering a wide range of disease severity showed sensitivities of 0.98 and 1.00 calculated at the patient level. This suggests that the two methods show no difference in their ability to guide surgeons in removing HPGs. This result was substantiated by the non-inferiority analysis.

The patient-level analysis yielded higher sensitivities than did the gland-level analysis. At the gland level, each gland must be identified separately in order to count as a “true positive”, which causes decreased sensitivity. By increasing the number of observations (glands) per patient, the likelihood of missing a gland in an individual increases, whereas the diagnosis of “any” gland in the patient tends to be fairly straightforward. Analogously, specificities will increase because at the gland level, multiple glands without disease are added to the data set, increasing the likelihood of calling a true-negative observation.

The study was carried out with two highly experienced specialists in parathyroid nuclear medicine. The Di-SPECT scan is known to be quite robust and is applicable for medium-trained readers. Determining whether the Choline PET scan has the same robustness calls for testing by inter- and intra-observer variability studies.

The surgeons were not blinded to the Choline PET/CT scan results, which creates a risk of bias. This was, however, the decision of the study group as it was considered unethical to withhold potentially valuable information from the surgeons. The study used only the surgeons’ report of the location of HPGs as “the truth”.

Disease severity varied widely, as the included subjects ranged from asymptomatic individuals diagnosed only by chance to patients with severely symptomatic PHPT. This may have made imaging more challenging, though equally so, as patients underwent both scans within a short timeframe.

## 5. Conclusions

In this prospective, blinded, clinical trial, the overall conclusion is that the choline PET method for preoperative detection of HPGs is not inferior to the generally accepted MIBI-SPECT wash-out method or the even more sensitive method of dual-isotope subtraction scintigraphy with ^99m^Tc-MIBI/^123^Iodide using SPECT/CT and planar pinhole imaging. The methods have considerably high sensitivities. However, the clinical implications of the two studies require further investigation. Moreover, whether the methods are interchangeable or supplementary to one another has yet to be determined.

## Figures and Tables

**Figure 1 diagnostics-10-00975-f001:**
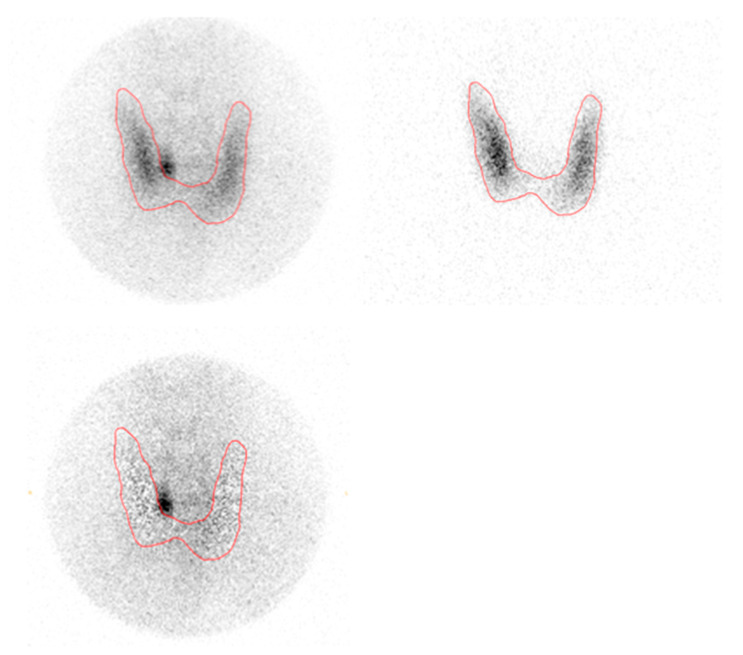
Dual isotope planar pinhole subtraction scintigraphy. Upper left: ^99m^Tc-sestamibi pinhole. Upper right: ^123^I pinhole. Bottom left: Pinhole subtraction.

**Figure 2 diagnostics-10-00975-f002:**
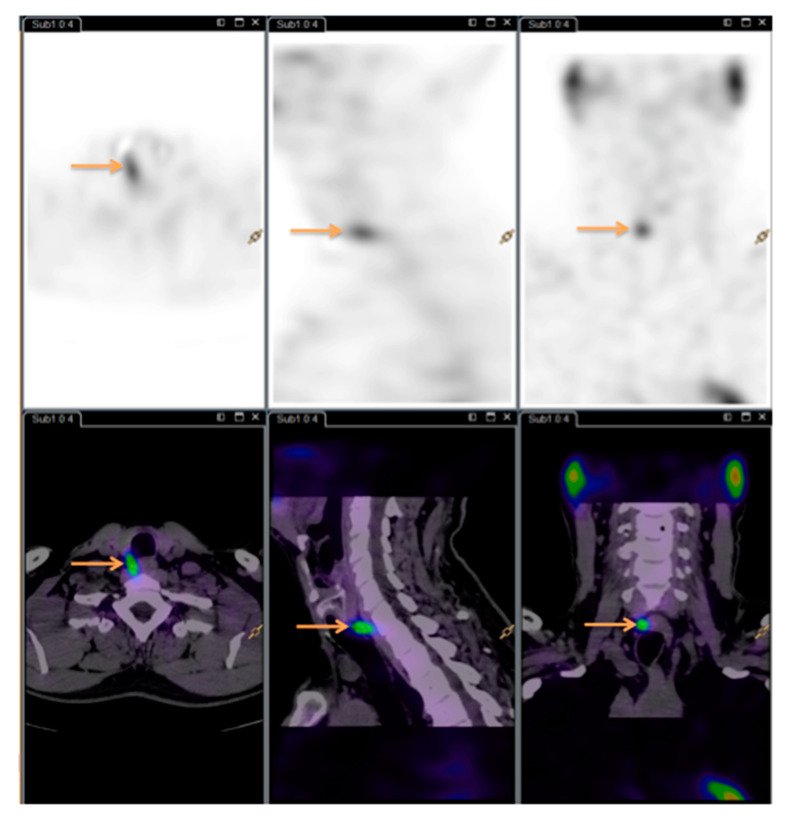
Dual isotope subtraction single-photon emission computed tomography/computed tomography (SPECT/CT) image. Arrows showing the hyperfunctioning parathyroid gland.

**Figure 3 diagnostics-10-00975-f003:**
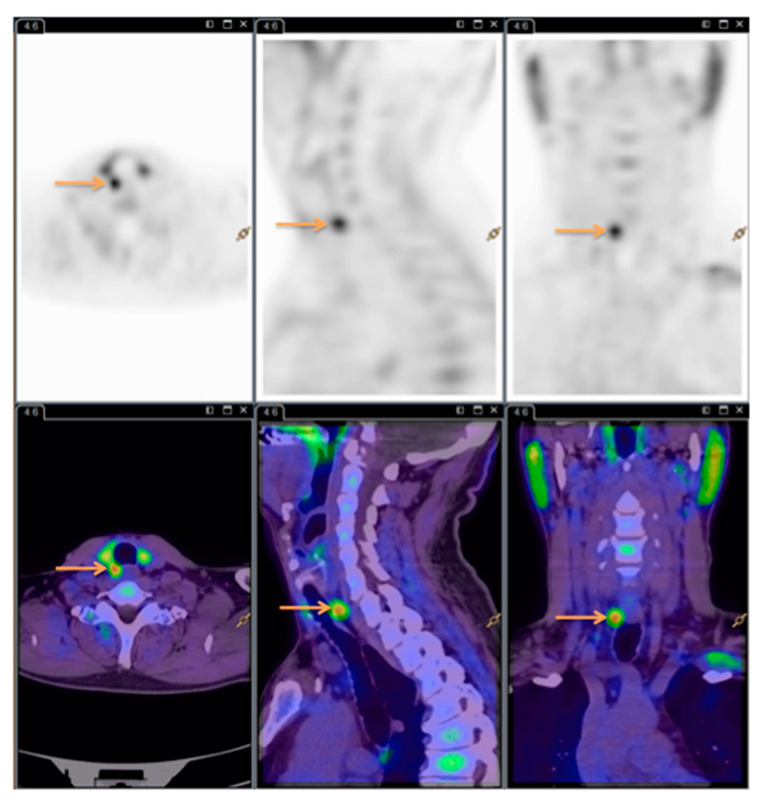
^11^C-Choline Positron emission tomography/computed tomography (PET/CT). Arrows showing the hyperfunctioning parathyroid gland.

**Table 1 diagnostics-10-00975-t001:** Baseline patient characteristics.

Number of patients, *n*		60	
Gender, *n* (%)	Female	45	(75%)
	Male	15	(25%)
		Median	Interquartile Range
Age (years)		62.2	(54.8–72.3)
Height (cm)		170	(165–178)
Weight (kg)		75	(65–84)
Preoperative plasma Ca^2+^ (mmol/L) *		1.41	(1.3–1.47)
Preoperative plasma PTH (pmol/L) ^∼^		13.0	(10.3–15.7)
Weeks between the two scans		0.9	(0.4–3.6)

* Upper reference limit 1.32 mmol/L. ^∼^ Upper reference limit 8.5 pmol/L.

**Table 2 diagnostics-10-00975-t002:** (**a**). Di-SPECT results as compared to the surgery report. (**b**). Choline PET results as compared to the surgery report.

**2a.**				
		Surgery Report		
		+	−	Total
Di-SPECT	+	52	4	56
	−	1	3	4
	Total	53	7	60
Sensitivity = 0.98 (95% CI: 0.90–1.00)				
**2b.**				
		Surgery Report		
		+	−	Total
Choline PET	+	53	6	59
	−	0	1	1
	Total	53	7	60
Sensitivity = 1.00 (95% CI: 0.93–1.00)				

**Table 3 diagnostics-10-00975-t003:** (**a**). Di-SPECT results as compared to the surgery report. (**b**). Choline PET results as compared to the surgery report.

**3a.**				
		Surgery Report		
		+	−	Total
Di-SPECT	+	59	4	63
	−	16	161	177
	Total	75	165	240
Sensitivity = 0.83 (95% CI: 0.72–0.91)				
**3b.**				
		Surgery Report		
		+	−	Total
Choline PET	+	60	4	64
	−	15	161	176
	Total	75	165	240
Sensitivity = 0.87 (95% CI: 0.76–0.93)				

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
