# Peer review of "11C-Choline PET/CT vs. 99mTc-MIBI/123Iodide Subtraction SPECT/CT for Preoperative Detection of Abnormal Parathyroid Glands in Primary Hyperparathyroidism: A Prospective, Single-Centre Clinical Trial in 60 Patients"

_diagnostics, 2020, doi:10.3390/diagnostics10110975_

Round 1
Reviewer 1 Report
In this manuscript, 11C-Choline PET/CT was evaluated in primary hyperparathyroidism patients, as a comparison with 99mTc-MIBI/123Iodide using SPECT/CT and planar pinhole imaging. As a result, 11C-Choline PET/CT showed promising value in locating hyperfunctioning glands. Overall, this manuscript is a good fit to publish on MDPI Diagnostics.
To improve the quality of this manuscript, the following comments should be considered in introduction and discussion.
Although 11C-Choline is an authentic compound and easy to make, the short half-life of 20 min allows maximum ~2 h shelf life, which generally requires onsite cyclotron and PET radiochemistry facility. From the clinical practice in our facility, we can only provide 4-7 11C-Choline doses for one production, and have to make 2-3 productions on the same day to meet clinical needs. This is inferior to its F-18 version radiotracers.
As the most recent advance, [18F]tetrafluoroborate (18F-TFB), the analog of iodide, was evaluated as highly efficient and sensitive radiotracer for Sodium/Iodide Symporter (NIS) PET imaging, showing very promising results in thyroid and NIS reporter gene imaging.
Major works on 18F-TFB are recommended to be cited:
1, Synthesis and biological evaluation of [ 18F]tetrafluoroborate: A PET imaging agent for thyroid disease and reporter gene imaging of the sodium/iodide symporter
November 2010 European Journal of Nuclear Medicine 37(11):2108-16
DOI: 10.1007/s00259-010-1523-0
2, Synthesis of 18F-Tetrafluoroborate (18F-TFB) via Radiofluorination of Boron Trifluoride and Evaluation in a Murine C6-Glioma Tumor Model
First published April 21, 2016, J Nucl Med . 2016 vol. 57 no. 9 1454-1459
doi: 10.2967/jnumed.115.170894
3, Safety, pharmacokinetics, metabolism and radiation dosimetry of 18F-tetrafluoroborate (18F-TFB) in healthy human subjects
EJNMMI Res 7, 90 (2017). https://doi.org/10.1186/s13550-017-0337-5
4, 18F-Tetrafluoroborate, a PET Probe for Imaging Sodium/Iodide Symporter Expression: Whole-Body Biodistribution, Safety, and Radiation Dosimetry in Thyroid Cancer Patients
J Nucl Med. 2017 Oct;58(10):1666-1671. doi: 10.2967/jnumed.117.192252
5, [18 F]Tetrafluoroborate ([ 18 F]TFB) and its analogs for PET imaging of the sodium/iodide symporter. Theranostics. 2018 Jun 24;8(14):3918-3931. doi: 10.7150/thno.24997. eCollection 2018.
6, Initial Clinical Investigation of [18F]Tetrafluoroborate PET/CT in Comparison to [124I]Iodine PET/CT for Imaging Thyroid Cancer.
Clinical Nuclear Medicine: March 2018 - Volume 43 - Issue 3 - p 162-167
doi: 10.1097/RLU.0000000000001977
Reviewer 2 Report
Well designed study. It would be of interest to readers and adds to the evidence in literature regarding role of PET-CT in hyperparathyroidism.
